# Composition and thermal processing evaluation of yeast ingredients as thiamin sources compared to a standard vitamin premix for canned cat food

Amanda N. Dainton[1], Markus F. Miller, Jr.[2‡], Brittany White[2‡], Leah Lambrakis[2‡], Charles Gregory Aldrich[1]*

1 Department of Grain Science & Industry, Kansas State University, Manhattan, KS, United States of America, 2 Simmons Pet Food, Inc, Siloam Springs, AR, United States of America

☯ These authors contributed equally to this work.
‡ MFM, BW and LL also contributed equally to this work.
* aldrich4@ksu.edu

**Data Availability Statement:** All relevant data are within the paper and/or Supporting Information files.

## Abstract

Significant improvement in thiamin retention of canned cat food has not been achieved by altering processing conditions. Some ingredients, such as yeasts, may supply thiamin able to withstand thermal processing. Therefore, the study objective was to evaluate yeast ingredients as thiamin sources for canned cat food. Six yeast ingredients were screened for thiamin content, and values ranged from 9.9–4,283.8 mg/kg dry matter basis (DMB). Treatments for thermal processing were arranged as a 2×4 factorial with 2 levels of vitamin premix (with or without) and 4 yeast ingredients (NY = none and LBV, BY, or EA from the ingredient screening). Replicates (n = 3) were processed in a horizontal still retort to an average lethality of 79.23 minutes. Thiamin degradation was analyzed as a mixed model with pre-retort thiamin content as a covariate and production day as a random effect. Main effects of vitamin premix and yeast and their interaction were significant at $P$-values less than 0.05. The Fisher's LSD post hoc comparison test was used to separate means. On average, experimental formulas retained 33.75% thiamin. The main effect of vitamin premix (average -42.9 mg/kg DMB) was not significant ($P > 0.05$). Thiamin degradation between NY (-31.3 mg/kg DMB) and BY (-33.8 mg/kg DMB) was similar ($P > 0.05$) whereas EA (-40.5 mg/kg DMB) and LBV (-55.6 mg/kg DMB) lost more ($P < 0.05$) thiamin than NY. The experimental formula of EA with vitamin premix (-70.3 mg/kg DMB) lost more ($P < 0.05$) thiamin than no yeast, BY, or EA without vitamin premix (average -17.4 mg/kg DMB) and all others (average -57.3 mg/kg DMB) were intermediate ($P > 0.05$). In summary, thiamin from yeast ingredients didn't exhibit better thermal stability than thiamin mononitrate. However, those ingredients with similar degradation levels or uniquely high thiamin levels may provide added value.

**Funding:** Ingredient and analytical costs were supported by Simmons Pet Food, Inc. The funder provided support in the form of salaries for authors M.F.J. Jr., B.W., and L.L. during the research experiment and for A.N.D. during an internship prior to the research experiment. The funder did not have any additional role in the study design, data collection and analysis, decision to publish, or preparation of the manuscript. The specific roles of these authors are articulated in the 'author contributions' section. There was no additional external funding for this study.

**Competing interests:** A.N.D. was employed by Simmons Pet Food, Inc. as a paid intern prior to conducting this research project. M.F.M. Jr., B.W., and L.L. were employed by Simmons Pet Food, Inc. while this research project was conducted. Simmons Pet Food, Inc. manufactures canned foods for dogs and cats. This does not alter our adherence to PLOS ONE policies on sharing data and materials. C.G.A. declares no competing interests.

# Introduction

Cats and other carnivores have a high requirement for thiamin [1] and consumption of a deficient diet can result in paralysis and death within a few weeks [2, 3]. Other observed symptoms of deficiency include impaired learning in young cats [4], low blood levels of transketolase [5], and brain lesions visible with magnetic resonance imaging [6, 7]. Reports in the literature indicate this is a widespread and common problem; a survey of 90 commercially available canned cat foods found that 15.6% of tested products contained less than 5.6 mg/kg thiamin on a dry matter basis (DMB) [8], which is the Association of American Feed Control Officials' (AAFCO) minimum recommended allowance for maintenance of adult cats [9]. Recalls due to insufficient thiamin have affected prominent companies in the pet food industry. Thus, this is a real and pervasive problem and not simply a function of naive or haphazard production practices.

Thiamin molecules (chemical formula: $C_{12}H_{17}N_4OS^+$) consist of a thiazole and a pyrimidine connected with a methylene bridge. The methylene bridge between the two portions of the thiamin molecule is relatively weak and can be destroyed during food processing [10]. Increased temperature and greater processing time decreased retention of thiamin in foods for human consumption [11–13]. These conditions are utilized to achieve commercial sterility, which is required by federal regulations in the United States. Commercial sterility is defined as the condition achieved by the application of heat wherein microorganisms able to reproduce when food is stored in ambient conditions and microorganisms and spores with a public health concern are not present in the food [14]. Foods that control for water activity in addition to the application of heat do not need to consider microorganisms and spores with a public health concern. Most canned pet foods do not control for water activity, so both categories of microorganisms and spores must be taken into account when processing canned pet foods. Therefore, altering processing conditions to minimize thiamin degradation to a significant and meaningful level is unlikely. This leaves the ingredient composition of canned cat food as another avenue.

Thiamin is typically supplied as thiamin mononitrate in a vitamin premix and/or as a separate ingredient. However, other ingredients may provide thiamin in different forms. One example is *Saccharomyces cerevisiae*, a common yeast already used in commercial pet food. Its use as a protein source and as a palatability enhancer in foods for companion animals is documented [15, 16]. In addition, yeast cells have a mechanism to bind thiamin for transport. Researchers isolated binding proteins from young *Saccharomyces cerevisiae* cells that were able to bind 63.4 picomoles of thiamin per milligram of protein at pH 5.5 [17]. Additionally, genetic analysis of *Saccharomyces cerevisiae* strains used for production of ethanol has identified amplified genes responsible for biosynthesis of thiamin [18]. However, this ingredient category has never been evaluated as a source of thiamin that might provide fortification through the retort process.

The objectives of this experiment were to identify yeast-based ingredients which might function as thiamin sources and to evaluate their effect on thiamin degradation in canned cat food after thermal processing. The hypothesis was that yeast-based sources of thiamin would retain more thiamin after thermal processing compared to thiamin mononitrate.

# Materials and methods

## Yeast ingredient screening

Six commercial dried yeasts were sourced from companies supplying ingredients to the pet food industry. They could be classified as two brewer's yeasts, one combination of an active

yeast and a brewer's yeast, one combination of a yeast extract and a brewer's yeast, one *Saccharomyces cerevisiae* fermentation product (FP; Diamond V, Cedar Rapids, IA), and one fortified inactive yeast (LBV, Lalmin B-Complex Vitamins; Lallemand Bio-Ingredients, a division of Lallemand Inc., Montréal, QC, CA). The two brewer's yeasts (BY and EA, respectively) were product #1064B (The Peterson Company, Kalamazoo, MI) and BGYADVANTAGE (The F.L. Emmert Company, Cincinnati, OH). The ingredient including an active yeast and a brewer's yeast (NS; NUCLEO-SACC) and the ingredient combining a yeast extract and a brewer's yeast (YS; YEA-SACC1026 OA) were both sourced from the same company (Alltech Inc., Nicholasville, KY). Five lots, except for LBV (n = 4), were collected for all ingredients. Each lot was analyzed for chemical composition in duplicate (Table 1). Three dried yeasts were selected from these six for evaluation in a canned cat food. Thiamin content was the primary selection criterion, with higher thiamin contents preferred and more likely to meet the minimum recommended amount of 5.6 mg thiamin per dry matter kg of processed canned cat food [9].

## Experimental treatment production

Eight formulas were created to test the retention of a standard vitamin premix containing thiamin mononitrate as the main thiamin source separate from and in addition to the yeast ingredients LBV, BY, and EA (Table 2). The experimental treatments were arranged as a 2×4 factorial with 2 categorical levels of vitamin premix (without vitamin premix or with vitamin premix) and 4 categorical levels of dried yeast (no yeast, LBV, BY, or EA). The formulas with vitamin premix included a vitamin premix at 0.08% of the total mass. This vitamin premix

**Table 1. Nutritional composition (mean ± standard deviation) of commercial yeast ingredients[1] selected as potential sources of protected thiamin for canned cat food.**

| Nutrient | NS | YS | FP | EA | LBV | BY |
|---|---|---|---|---|---|---|
| Lots analyzed, n | 5 | 5 | 5 | 5 | 4 | 5 |
| Moisture, % | 6.9 ± 0.21 | 4.5 ± 0.12 | 8.3 ± 0.15 | 4.3 ± 0.28 | 5.0 ± 0.41 | 7.4 ± 0.81 |
| | ---------------------------------------------------------------Dry matter basis---------------------------------------------------------------- | | | | | |
| Thiamin, mg/kg | 13.9 ± 2.58 | 10.6 ± 2.07 | 9.9 ± 0.85 | 12.0 ± 1.59 | 4283.8 ± 176.18 | 36.8 ± 5.97 |
| Crude protein, % | 45.6 ± 0.34 | 23.4 ± 1.04 | 11.8 ± 0.66 | 54.5 ± 1.09 | 57.3 ± 1.29 | 48.6 ± 4.34 |
| Crude fat, % | 0.41 ± 0.390 | 2.74 ± 0.292 | 0.69 ± 0.251 | 1.68 ± 0.256 | 2.50 ± 0.286 | 0.34 ± 0.207 |
| Crude fiber, % | 0.35 ± 0.130 | 4.59 ± 1.775 | 38.52 ± 2.261 | 3.65 ± 0.683 | 0.38 ± 0.150 | 0.50 ± 0.556 |
| Ash, % | 5.87 ± 0.132 | 41.25 ± 0.776 | 7.90 ± 0.465 | 7.08 ± 0.221 | 6.34 ± 0.202 | 6.35 ± 0.945 |
| NFE[2], % | 47.8 ± 0.47 | 28.0 ± 2.13 | 41.1 ± 1.30 | 33.1 ± 0.60 | 33.5 ± 1.10 | 44.2 ± 5.35 |
| Calcium, % | 0.24 ± 0.044 | 14.64 ± 0.497 | 0.64 ± 0.056 | 0.49 ± 0.029 | 0.23 ± 0.013 | 0.33 ± 0.028 |
| Phosphorus, % | 1.480 ± 0.0387 | 0.952 ± 0.0313 | 0.324 ± 0.0422 | 0.782 ± 0.0271 | 1.093 ± 0.1007 | 1.434 ± 0.1747 |
| Potassium, % | 1.710 ± 0.0615 | 0.430 ± 0.0247 | 1.865 ± 0.0925 | 2.337 ± 0.0553 | 2.546 ± 0.0759 | 1.459 ± 0.3053 |
| Sodium, % | 0.020 ± 0.0000 | 0.020 ± 0.0000 | 0.767 ± 0.1178 | 0.017 ± 0.0067 | 0.095 ± 0.0173 | 0.101 ± 0.1422 |
| Magnesium, % | 0.196 ± 0.0101 | 1.629 ± 0.0722 | 0.427 ± 0.0507 | 0.308 ± 0.0076 | 0.118 ± 0.0089 | 0.195 ± 0.0202 |
| Sulfur, % | 0.43 ± 0.013 | 0.25 ± 0.005 | 0.31 ± 0.034 | 0.41 ± 0.011 | 0.54 ± 0.044 | 0.39 ± 0.040 |
| Iron, mg/kg | 68 ± 6.3 | 1627 ± 64.7 | 156 ± 23.6 | 112 ± 8.1 | 47 ± 3.1 | 175 ± 64.2 |
| Copper, mg/kg | 29.49 ± 1.906 | 8.65 ± 0.788 | 4.01 ± 0.558 | 16.96 ± 1.067 | 5.50 ± 0.265 | 6.01 ± 2.479 |
| Manganese, mg/kg | 9.32 ±1.563 | 146.70 ± 8.808 | 20.40 ± 4.518 | 38.12 ± 1.943 | 10.25 ± 0.612 | 9.24 ± 2.477 |
| Zinc, mg/kg | 82 ± 3.3 | 105 ± 6.1 | 68 ± 15.4 | 60 ± 7.0 | 88 ± 9.2 | 208 ± 103.3 |

[1] NS = NUCLEO-SACC; YS = YEA-SACC; FP = *Saccharomyces cerevisiae* fermentation product; EA = BGYADVANTAGE; LBV = Lalmin B-Complex Vitamins; BY = spray-dried brewer's yeast #1064B.

[2] NFE = nitrogen free extract, calculated (Dry matter basis contents of crude protein, crude fat, crude fiber, and ash subtracted from 100).

**Table 2. Ingredient composition of canned cat foods containing different levels of a vitamin premix and/or a dried yeast ingredient[1].**

| Ingredient, % as-is | No vitamin premix | | | | Contains vitamin premix | | | |
|---|---|---|---|---|---|---|---|---|
| | NY | LBV | BY | EA | NY | LBV | BY | EA |
| Basal batter[2] | 94.92 | 94.92 | 94.92 | 94.92 | 94.92 | 94.92 | 94.92 | 94.92 |
| Ground brewer's rice | 5.08 | 4.43 | 0.08 | 0.08 | 5.00 | 4.35 | - | - |
| Vitamin premix | - | - | - | - | 0.08 | 0.08 | 0.08 | 0.08 |
| Lallemand yeast | - | 0.65 | - | - | - | 0.65 | - | - |
| Peterson yeast | - | - | 5.00 | - | - | - | 5.00 | - |
| Emmert yeast | - | - | - | 5.00 | - | - | - | 5.00 |

[1] NY = no yeast; LBV = Lalmin B-Complex Vitamins; BY = spray-dried brewer's yeast #1064B; EA = BGYADVANTAGE.

[2] 1 kg of basal batter contains 283.48 g mechanically deboned low ash chicken, 230.33 g pork liver, 223.59 g water, 148.83 g ground chicken, 106.30 g steam, 3.19 g guar gum, 1.35 g potassium chloride, 1.06 g mineral premix for cats, 0.74 g kappa carrageenan, 0.57 g taurine, 0.53 g salt, and 0.02g 50% vitamin E.

contained thiamin mononitrate as the thiamin source. The 0.08% inclusion was chosen to emulate a commercial canned cat food providing roughly 2,343% of the AAFCO minimum recommended level of 5.6 mg thiamin per kg of diet dry matter [9]. The level of LBV (0.65%) was chosen to match this value based on the average dry matter thiamin content from the initial ingredient screening. The levels for BY and EA were capped at 5% even though this level was not expected to provide the AAFCO minimum recommended level for thiamin after thermal processing. This maximum level was chosen to ensure that all experimental treatments were practical from a processing and ingredient formulation perspective.

Formulas were produced once per replicate day of processing (n = 3). Production of a 1,361 kg batch of basal batter began by grinding (Scansteel Foodtech, Denmark) mechanically deboned low ash chicken (Simmons Animal Nutrition, Siloam Springs, AR), ground chicken (Simmons Animal Nutrition, Siloam Springs, AR), and pork liver (BHJ USA, LLC., Omaha, NE) through a die plate with 6.35 mm openings. Next, the ground meats were mixed (MTB-40-100P; MTC, Temple, TX) with a gravy prepared prior with a triblender (F3218; Alfa Laval Inc., Richmond, VA). This gravy contained the water, guar gum (Tilley Chemical Company Inc., Baltimore, MD), potassium chloride (Bill Barr & Company, Overland Park, KS), mineral premix (Trouw Nutrition USA LLC, Highland, IL), kappa carrageenan (Mannasol Products Ltd., Cheshire, United Kingdom), taurine (Avid Organics, Gujarat, India), salt (Compass Minerals America, Overland Park, KS), and 50% vitamin E (DSM Nutritional Products, Heerlen, NL) for the formula. Steam was added while mixing to increase moisture content and to raise the temperature of the batter to 50˚C. The batter was processed through a three-plate emulsifier (Comvair-149.14 kW; Reiser, Canton, MA) and 91 kg of batter was sub-sampled to create the eight canned pet foods. Each formula was produced by adding the ground brewer's rice (Les Aliments Dainty Foods, Windson, ON, Canada), vitamin premix (Nutra Blend LLC, Neosho, MO), and/or yeast (depending on the treatment) directly to 7.6 kg of basal batter. Batter temperature was maintained at a minimum of 50˚C with a hot plate. The addition was done while mixing the batter with a power drill (#DR560; Black + Decker, Towson, MD) equipped with a paint mixing attachment for 1 minute to minimize clumping and to ensure a complete distribution of the added dry ingredients. A sample of each batter was collected and frozen for chemical analysis before cans (size 307×109; Crown Holdings, Philadelphia, PA) were filled with 156 g of batter and seamed (Pneumatic Scale Angelus, Stow, OH) with an easy-open lid.

A minimum of 1 can per formula replicate contained a thermocouple (Ecklund-Harrison Technologies Inc., Fort Myers, FL) for thermal process validation and cook value calculations. Lethality and cook value ($C_{100}$) were calculated using Eq 1 [19] and Eq 2 respectively, wherein

$T_C(t)$ is the temperature recorded by the thermocouple at time t and $\Delta t$ represents the length of time between temperature measurements (15 seconds or 0.25 minutes). The temperature of 121.11°C and z-value of 10°C for the calculation of lethality are reference values for *Clostridium botulinum*. This hardy bacterium poses a public health concern for thermally processed low-acid foods [20]. Reference values for $C_{100}$ (temperature = 100°C; z-value = 10°C) were derived from experiments with thiamin because the nutrient is highly sensitive to thermal processing. The units for lethality and $C_{100}$ are relative time (minutes) the food product could have been processed for at the respective reference temperature. Both integral equations were solved using the trapezoid rule (Eqs 3 and 4, respectively). All cans for each production day were processed in a horizontal steam batch retort (Versatort Multimode 1520; Allpax, Covington, LA) under a process schedule designed to mimic a worst-case scenario in production with a 10 minute come-up cycle, a 63 minute cooking cycle (target temperature = 123°C), and a 27 minute cooling cycle. Afterwards, cans were removed from the retort and cooled to room temperature before analysis.

$$Lethality = \int 10^{\frac{T_C(t)-121.11°C}{10°C}}\Delta t \qquad (1)$$

$$C_{100} = \int 10^{\frac{T_C(t)-100°C}{33°C}}\Delta t \qquad (2)$$

$$Lethality = \sum_0^t 10^{\frac{T_C(t)-121.11°C}{10}}\Delta t \qquad (3)$$

$$C_{100} = \sum_0^t 10^{\frac{T_C(t)-100°C}{33}}\Delta t \qquad (4)$$

## Chemical analyses

All samples were analyzed by a commercial laboratory (Midwest Laboratories; Omaha, NE) for moisture (AOAC 930.15), thiamin (AOAC 942.23), crude protein (AOAC 99003), crude fat (AOAC 2003.05 for dry samples, AOAC 954.02 for wet samples), crude fiber (AOCS Ba 6a-05), ash (AOAC 942.05), sulfur (AOAC 985.01), phosphorus (AOAC 985.01), potassium (AOAC 985.01), magnesium (AOAC 985.01), calcium (AOAC 985.01), sodium (AOAC 985.01), iron (AOAC 985.01), manganese (AOAC 985.01), copper (AOAC 985.01), and zinc (AOAC 985.01). Results below the minimum detection level were treated as zeros. Mechanically deboned low ash chicken, pork liver, and ground chicken were not analyzed for crude fiber content. Samples of the post-retort diets were generated by compositing three cans from the individual replicates. All analyses were conducted in duplicate. Nitrogen free extract (NFE) was calculated by subtracting the DMB contents of crude protein, crude fat, crude fiber, and ash from 100%.

## Statistical analysis

Data from the initial screening of dried yeasts were presented as mean values across analyzed lots with their corresponding standard deviation.

Initial internal can temperature, lethality, and $C_{100}$ were analyzed as a 2-way analysis of variance (ANOVA) with vitamin premix and yeast as fixed effects and production day as a

random effect. The procedure GLIMMIX was used for this analysis (SAS v. 9.4; SAS Institute, Cary, NC). Nutrient contents of pre-retort batters and processed diets were presented as mean values ± standard deviation. Thiamin content of individual formulas was compared to 5.6 mg/kg DMB [9], the minimum recommended value for maintenance of adult cats [9], using a single-sample lower-tail t test. Dietary thiamin content was considered less than the minimum recommended value if the *P*-value was less than 0.05.

Change in thiamin content was calculated by subtracting pre-retort batter thiamin content from processed diet thiamin content. Data were transformed [wherein "*x*" represents a data point: $(x* - 1)^{1/3}$] to meet the model assumptions of equal variance and normality for statistical analysis. Significance of the main effects for vitamin premix and yeast and their interaction were determined with a 2-way analysis of covariance (ANCOVA). Pre-retort batter thiamin content was included as a covariate and production day as a random blocking factor. Retention was analyzed with this methodology instead of as a percent loss or percent retention to maximize the stability of the response variable and to account for the variation due to initial thiamin content [21]. Denominator degrees of freedom were corrected using the Kenward-Roger adjustment. Individual means were separated by Fisher's LSD and significance was set at α = 0.05. The procedure GLIMMIX was employed in this analysis (SAS v. 9.4; SAS Institute, Cary, NC). Data were presented as means with a 95% confidence interval (CI) in pre-transformation units.

## Results

### Nutrient composition of screened yeast ingredients

Differences in nutrient composition were observed in the initial screening of dried yeast ingredients (Table 1). Thiamin content was highest for LBV (4283.8 mg/kg DMB) with all other ingredients similar to each other (9.9–36.8 mg/kg DMB). Ranges for crude protein (from 11.8% DMB in FP to 57.3% DMB in LBV), crude fiber (from 0.35% DMB in NS to 38.52% DMB in FP), and ash (from 5.87% DMB in NS to 41.25% DMB in YS) were also wide. Crude fat content was less varied with the highest contents measured in LBV and YS (average = 2.62% DMB), lowest in NS, FP and BY (average = 0.48% DMB), and intermediate in EA (1.68% DMB). Notable findings for the mineral concentrations included higher levels of calcium, iron, and manganese in YS (14.640% DMB, 1627.0 mg/kg DMB, and 146.70 mg/kg DMB, respectively) and a higher level of zinc in BY (207.5 mg/kg DMB).

### Nutrient content of raw ingredients, pre-retort batters, and processed diets

Nutrient composition varied among ingredients used to create the experimental formulas (S1 Table). Ingredients in the basal batter contributed low levels of thiamin (average 4.3 mg/kg DMB). The ingredient with the highest thiamin content was the vitamin premix at 17933.3 mg/kg DMB. Brewer's rice contained low levels (1.6 mg/kg DMB) consistent with the ingredients included in the basal batter. On average, the three yeasts selected contained 515% more crude protein, 84.8% less crude fat, 95.0% less crude fiber, 78.5% less ash, and 55.4% more NFE than the vitamin premix. The BY and EA yeasts contained similar levels of thiamin, averaging 27.5 mg/kg DMB. While LBV contained more thiamin than BY and EA, it still only contained 7.09% of the thiamin provided in the vitamin premix. Mean values for crude protein, crude fat, calcium, phosphorus, potassium, sodium, magnesium, iron, copper, manganese, and zinc for all batters and diets were above the AAFCO recommended minimum levels for adult cats (S2 Table). There are no minimum recommendations for moisture, crude fiber, ash, or NFE. Moisture content averaged 82.0 ± 1.50% across the diets pre- and post-retort processing

(Table 3). Thiamin content was lower for all diets after retort processing. The post-retort diet that did not contain the vitamin premix or a yeast ingredient (0.7 ± 1.02 mg/kg DMB) and the post-retort diet including EA without the vitamin premix (2.2 ± 0.37 mg/kg DMB) did not meet the minimum recommended level of 5.6 mg/kg DMB thiamin for adult cats ($P = 0.0070$ and $P = 0.0042$, respectively). All other post-retort diets contained enough thiamin to meet or surpass the minimum recommended level ($P > 0.05$).

## Thiamin degradation due to thermal processing

At least 1 thermocouple was successful for all formula replicates, except for the formula containing BY and no vitamin premix on production day 3. No main effects of vitamin premix or yeast or their interaction were detected ($P > 0.05$) for thermal processing data. Initial internal can temperature averaged 30.01 ± 4.471˚C and total lethality and $C_{100}$ averaged 79.23 ± 7.418 and 280.76 ± 17.750 minutes, respectively.

Inclusion of the vitamin premix (average -62.8 mg/kg DMB; 95% CI = -97.4 mg/kg DMB, -37.6 mg/kg DMB) did not affect ($P > 0.05$; Fig 1) the change in thiamin content compared to formulas without the vitamin premix (-23.0 mg/kg DMB; 95% CI -41.7 mg/kg DMB, -10.9 mg/kg DMB). This represented an average 33.75% retention of thiamin after thermal processing regardless of vitamin premix inclusion.

Change in thiamin content was affected ($P < 0.05$) by the main effect of yeast (Fig 2) and the interaction between yeast and vitamin premix (Fig 3). The greatest loss of thiamin was observed for the LBV yeast (average -55.6 mg/kg DMB; 95% CI -69.9 mg/kg DMB, -43.3 mg/kg DMB). More thiamin was lost with EA (average -40.5 mg/kg DMB; 95% CI -46.4 mg/kg DMB, -35.1 mg/kg DMB) vs. NY (average -31.3 mg/kg DMB; 95% CI -39.0 mg/kg DMB, -24.7 mg/kg DMB) with BY (average -33.8 mg/kg DMB; 95% CI -39.2 mg/kg DMB, -28.9 mg/kg DMB) not different from either. The formula including the vitamin premix and EA (average -70.3 mg/kg DMB) exhibited a greater loss in thiamin content compared to formulas without the vitamin premix and either EA, BY, or NY (average -17.4 mg/kg DMB). Formulas containing LBV with or without the vitamin premix and EA or NY with the vitamin premix (average -57.3 mg/kg DMB) were intermediate and not different from the others. If these changes are expressed as relative percent losses, the diets would be ranked as follows: BY without the vitamin premix (45.8%), NY with the vitamin premix (60.2%), BY with the vitamin premix (62.4%), LBV without the vitamin premix (63.2%), LBV with the vitamin premix (68.8%), EA with the vitamin premix (72.5%), EA without the vitamin premix (74.1%), and NY without the vitamin premix (82.9%).

**Table 3. Moisture and thiamin contents (mean ± standard deviation) of pre-retort and post-retort canned cat foods containing different levels of a vitamin premix and/or a yeast ingredient[1].**

| | No vitamin premix | | | | Contains vitamin premix | | | |
|---|---|---|---|---|---|---|---|---|
| Nutrient | NY | LBV | BY | EA | NY | LBV | BY | EA |
| Pre-retort moisture, % | 82.7 ± 0.34 | 79.7 ± 4.29 | 82.3 ± 0.25 | 81.7 ± 1.79 | 81.0 ± 0.88 | 81.3 ± 0.93 | 81.1 ± 0.60 | 82.7 ± 1.13 |
| Post-retort moisture, % | 82.9 ± 0.98 | 82.6 ± 1.24 | 83.1 ± 0.74 | 81.9 ± 1.09 | 82.7 ± 1.13 | 82.5 ± 1.32 | 82.7 ± 0.74 | 82.0 ± 1.08 |
| | -------------------------------------------------------------Dry matter basis------------------------------------------------------------- | | | | | | | |
| Pre-retort thiamin, mg/kg | 4.1 ± 1.38 | 54.1 ± 14.07 | 10.7 ± 1.40 | 8.5 ± 1.55 | 137.3 ± 15.95 | 191.5 ± 7.26 | 161.3 ± 21.23 | 169.2 ± 26.50 |
| Post-retort thiamin, mg/kg | 0.7* ± 1.02 | 19.9 ± 4.29 | 5.8 ± 0.54 | 2.2* ± 0.37 | 54.7 ± 18.64 | 59.8 ± 7.29 | 60.7 ± 11.04 | 46.5 ± 5.78 |

[1] NY = no yeast; LBV = Lalmin B-Complex Vitamins; BY = spray-dried brewer's yeast #1064B; EA = BGYADVANTAGE.

* Least square mean thiamin content is less than 5.6 mg/kg dry matter basis.

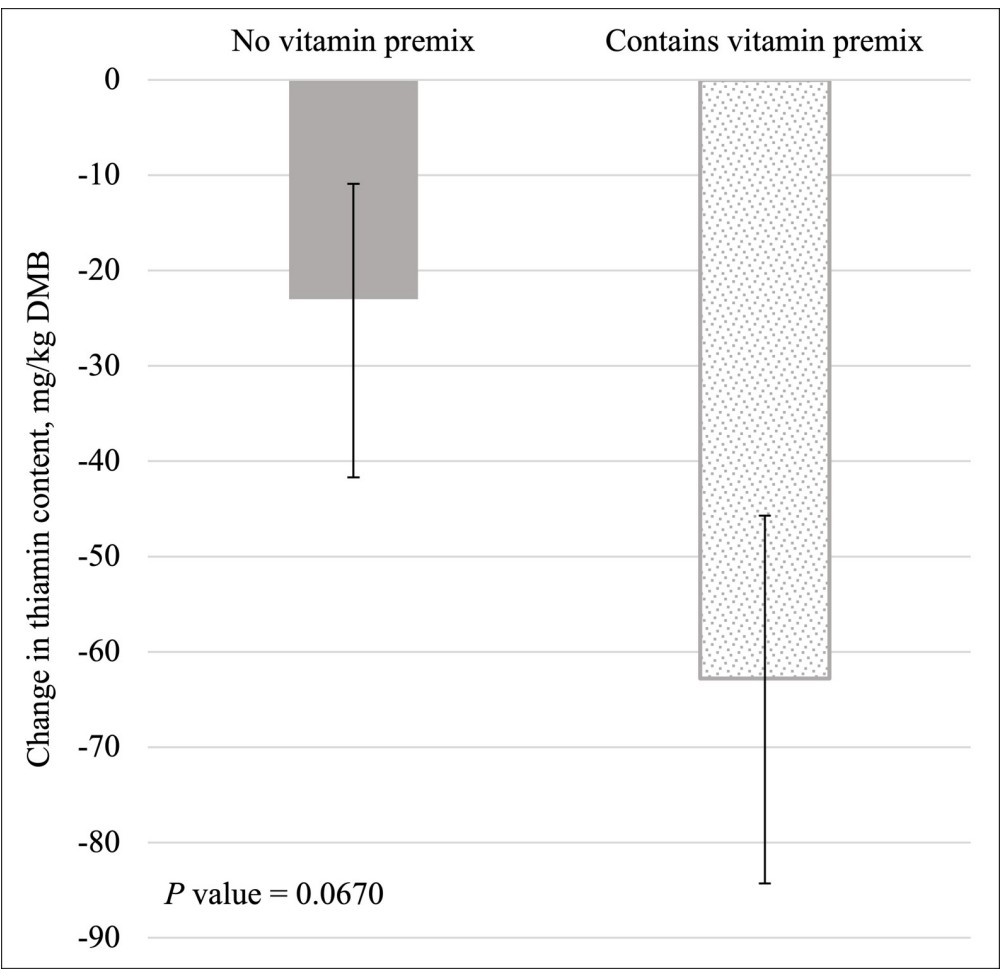

**Fig 1. Main effect of vitamin premix inclusion on the change in the dry matter basis (DMB) thiamin content (mean values with 95% confidence interval) of canned cat food.**

## Discussion

The aim of this work was to identify yeast ingredients as potential thiamin sources and to evaluate the stability of their intrinsic thiamin during thermal processing. Preliminary studies identified that retort processing conditions [22], package size and material [23], and protein source and the presence of sulfites [24] influenced thiamin content in canned cat food. However, none of those experiments identified a solution to minimize thiamin degradation that could be applied to all thermally processed cat foods. If a yeast ingredient with similar or better thermal resistance compared to thiamin mononitrate, then it could be suitable as a novel thiamin source.

### Nutrient composition of screened yeast ingredients

Dried yeast ingredients are commonly included in commercial pet foods as protein sources and dose-dependent palatability enhancers for cats [25, 26]. As such, reports of nutritional content of these ingredients are available for comparison. Moisture contents for the ingredients screened in the present experiment were similar and in-line with published values for dried brewer's yeasts and other yeast ingredients. Many manuscripts do not present thiamin

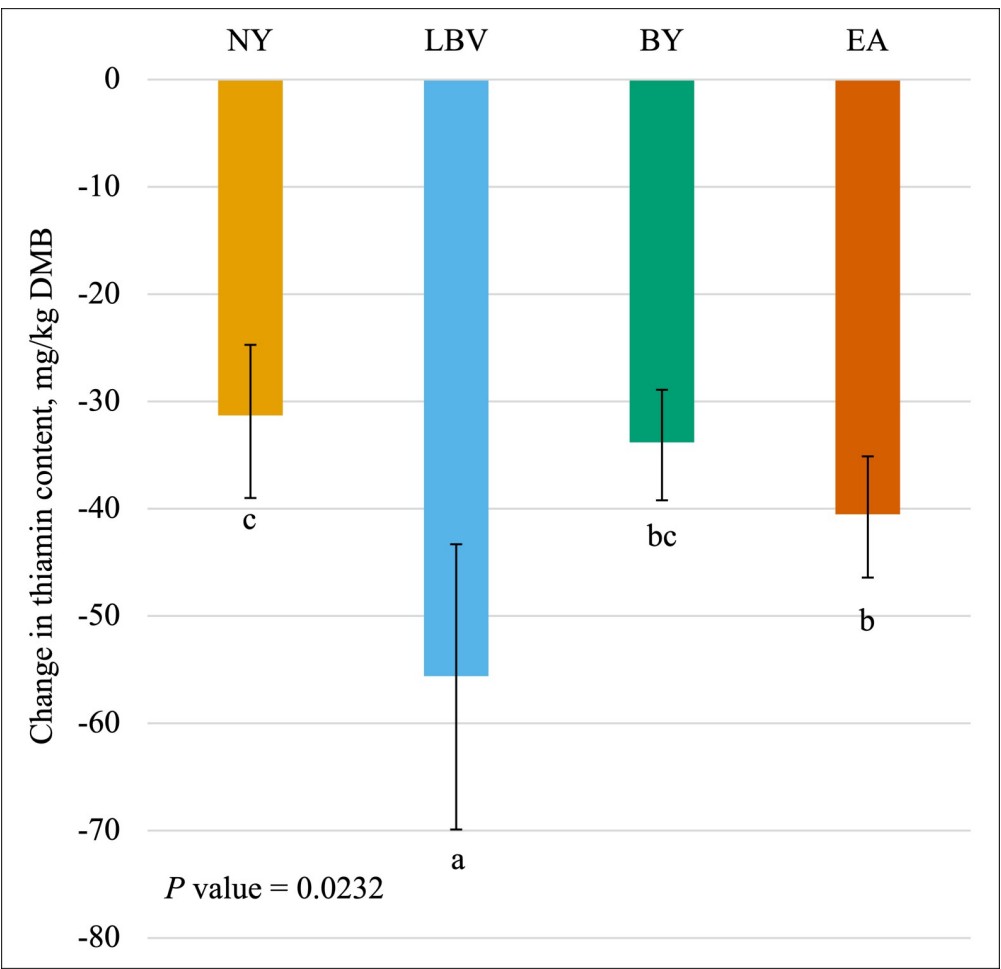

**Fig 2. Main effect of yeast inclusion on the change in the dry matter basis (DMB) thiamin content (mean values with 95% confidence interval) of canned cat food.** [abc] Means within the same chart that do not share a superscript are different ($P < 0.05$). [1] NY = no yeast; LBV = Lalmin B-complex vitamins; BY = spray-dried brewer's yeast #1064B; EA = BGYADVANTGE.

contents for yeast ingredients intended for animal consumption. Thiamin content of a dried brewer's torula yeast was 6.7 mg/kg DMB [1], which was lower than all six yeast ingredients screened in the present experiment. Crude protein and fat contents of all 6 yeasts fell in line with ingredients evaluated in animal feeding experiments [15, 27–29]. The FP ingredient contained notably high levels of crude fiber compared to the other yeast ingredients, which was expected. A *Saccharomyces cerevisiae* fermentation product fed to adult dogs similarly contained high levels of fiber [30]. Ash contents were similar in the screened yeast ingredients, with the exception of YS. The notably high ash content of YS was likely driven by its calcium and iron contents, which were also higher than the other screened yeast ingredients. However, there are no maximum calcium or iron levels for adult cats provided by AAFCO [9].

## Nutritional content of raw ingredients, pre-retort batters, and processed diets

The initial screening of commercially available yeasts yielded three ingredients for inclusion in canned cat foods for thermal processing. These yeasts were LBV, BY, and EA and their average

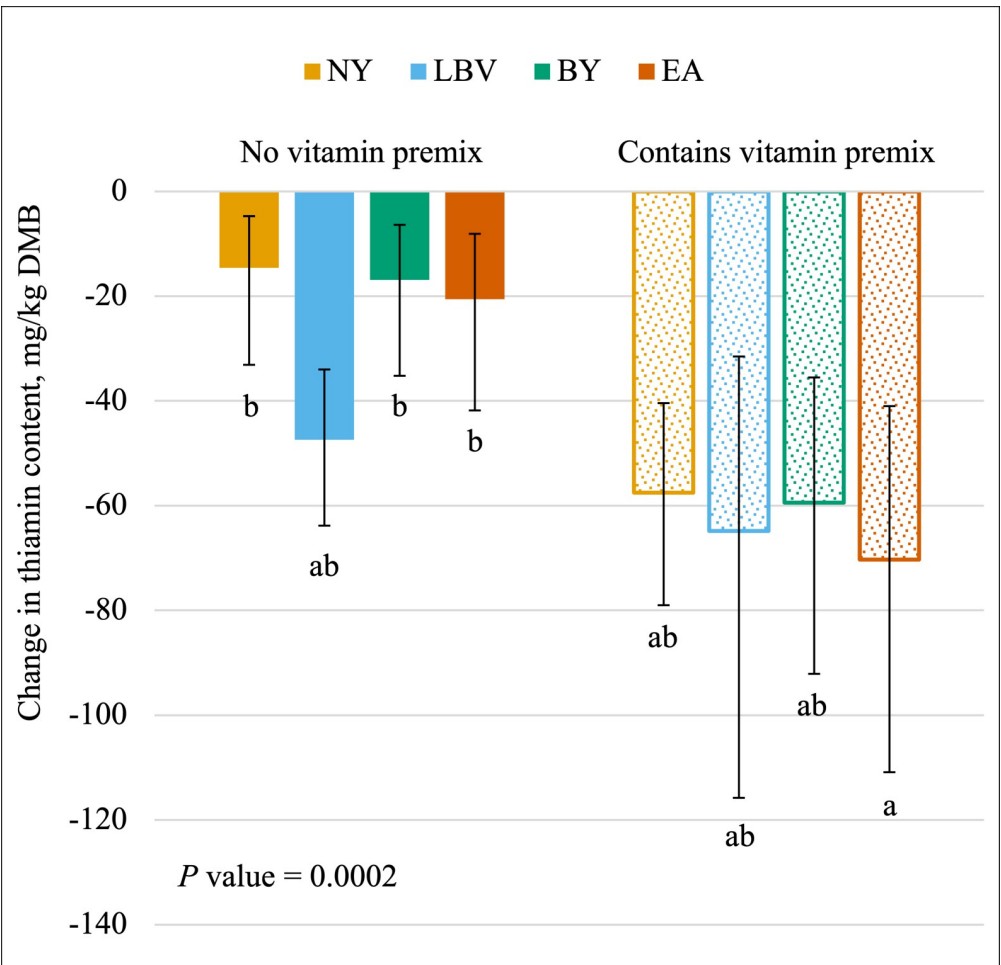

**Fig 3. The interaction of vitamin premix inclusion and yeast inclusion on the change in the dry matter basis (DMB) thiamin content (mean values with 95% confidence interval) of canned cat food.** [ab] Means within the same chart that do not share a superscript are different ($P < 0.05$). [1] NY = no yeast; LBV = Lalmin B-complex vitamins; BY = spray-dried brewer's yeast #1064B; EA = BGYADVANTGE.

DMB thiamin contents were 4283.8, 36.8, and 12.0 mg/kg, respectively, across multiple production lots. As detailed in the previous section, dried yeast ingredients have not been extensively evaluated for thiamin content. Limited reports suggested thiamin content of 5.1 mg/kg as-is basis in a brewer's yeast [31], 22.2 mg/kg DMB in a *Candida tropicalis* yeast biomass [32], and 6.7 mg/kg DMB in a torula brewer's yeast [1]. The wide range in thiamin contents observed suggested that one value cannot accurately describe all dried yeasts. Thiamin content should be analyzed when evaluating a new ingredient and source. Nevertheless, the dried yeasts selected for evaluation in thermally processed diets contained at least 100% more thiamin compared to published values for brewer's yeasts.

Analysis of the yeasts used for formula production found that LBV used in the formulas contained 70.3% less thiamin than the average screening lot. On the other hand, EA used in formula production contained 107.5% more thiamin. The BY ingredient was the most consistent and only contained 18.2% less thiamin. Storage can affect thiamin content in yeasts, with reported losses of 13–44% within 6 months of storage and 55–68% within 12 months [33]. As such, it is possible that LBV and BY supplied for experimental formula production were older

than the lots included in the initial screening. The higher thiamin content in EA used to produce formulas suggested that thiamin content in the ingredient was more variable than suggested by the screening. It is also possible that the differences in thiamin content between samples was influenced by the inherent variability in thiamin analysis or by sampling variation. Nevertheless, these differences led to lower-than-expected batter thiamin contents for those containing BY or LBV and higher than expected values for batters containing EA. Other nutrients were more consistent between the screening and production yeast samples. Supply chain management and accurate analytical data for these ingredients is critical if their intended purpose is to supply thiamin for commercial canned cat foods.

Nutrient content, especially thiamin, of ingredients used to produce commercial canned cat food is scarce in the literature. Data herein, especially the thiamin content of the pre-thermal processing batter that did not contain the vitamin premix or a yeast, suggested that thiamin content of these ingredients did not contribute to the overall thiamin content of the diet in a meaningful way. This information was still valuable for understanding the background thiamin content of the experimental formulas and can help pet food formulators when designing new products.

Ground brewer's rice was chosen as the space-filling ingredient for its low thiamin content and similar dry matter content compared to the vitamin premix and selected yeasts. For example, nutritional analysis of a whole brewer's rice yielded a moisture content of 11.00% and a thiamin content of 1.4 mg/kg as-is basis (or 1.6 mg/kg DMB) [1]. These values were comparable to the 12.5% moisture and 1.6 mg/kg DMB thiamin of the brewer's rice used in the present experiment. This led to similar moisture contents (average 82.0 ± 1.50%) across all 8 formulas. Typically, canned pâté style pet foods do not contain more than 78% moisture [34, 35]. As such, the formulas produced in this experiment may not fit the criteria of commercial pâté-style pet foods from that perspective. However, commercial canned pet foods that consist of a meat chunk in a liquid (colloquially called "chunks in gravy") may contain 82% moisture, which is very similar to the moisture content of the diets in the present experiment. The diets produced in the present experiment are still relevant to the pet food industry even though the moisture content was higher than anticipated. This could be modified by decreasing the amount of water and/or steam in the formula. Moisture content can influence the rate of heat penetration [36] but likely did not affect thiamin retention of the 8 formulas differently in the present experiment. Future experiments should more closely monitor steam and water addition to yield moisture contents less than 78% when producing pâté style canned pet foods.

Except for diets containing no vitamin premix with either NY or EA, all pre-retort batters and processed diets met the AAFCO nutrient recommendations for adult cats. As such, all diets except for the two previously mentioned represent potential commercial formulations from a macronutrient and mineral perspective. Further analysis to verify this would include amino acid and fatty acid contents, as crude protein and crude fat do not adequately describe these nutrients.

### Thiamin degradation due to thermal processing

Many factors affecting thermal processing and thiamin degradation have been identified and were controlled in the present experiment. Differences in moisture content and initial internal can temperature affect the kinetics of heat penetration [37, 38] and in turn influence thiamin losses due to thermal processing. However, diets in the present experiment had similar moisture contents and initial internal can temperatures. Preliminary research suggested that container type and size influence thiamin degradation [23], but the same container type and size was filled with the same amount of food in the present experiment. This suggested that differences

observed in thiamin loss in this experiment were due to ingredient inclusions and not influenced differently by processing conditions. Ingredients can affect thiamin degradation as well. The effect of thiaminase, an enzyme capable of destroying thiamin, was not considered because the meats were not different between diets and the enzyme is found in fish [39, 40], which were not included in the treatments. Additionally, none of the ingredients in the basal batter were preserved with sulfites, which are commonly used to preserve dehydrated potatoes and other carbohydrate visual inclusions [41]. Sulfites have been linked to thiamin degradation in thermally processed foods for humans [42] and pets [24]. Nevertheless, these situations illustrate that thiamin degradation can affected by the ingredients present in the formulation.

It is important to note that the scheduled process employed in the present experiment resulted in greater lethality than is normal for canned pet food. The typical lethality for commercial products was reported between 12 and 14 minutes [43]. This would result in a lower degree of thermal processing and less thiamin loss would likely be observed. Another experiment with a less harsh scheduled process would be necessary to determine the effect of the test ingredients on thiamin loss in a more typical production setting.

Average thiamin retention across all formulas was 33.75% with a range from to as described in the Results, which was lower than reported in a preliminary study at roughly 69% retention [23]. However, that experiment processed wet cat foods to a lethality of 8 minutes, which was less thermal processing than applied in this experiment. The Association of American Feed Control Officials gives a broad guidance of 90% degradation, or 10% retention, of thiamin due to retort processing [9]. However, this guidance does not consider the range of scheduled processes that can be employed to achieve commercial sterility. It is likely that the scheduled process and final lethality have a significant impact on thiamin retention. This was not a factor in the present experiment and should be confirmed in future research. The 33.75% thiamin retention observed herein was more in-line with approximately 30% thiamin retention for thermally processed beef liver [44] and 23.8–27.8% retained thiamin for pressure-cooked red-gram splits [13].

The cook value has been used to describe the degree of thermal processing on the quality of thermally processed food products, with higher cook values suggesting lower quality [45]. However, cook values in the present experiment were not different even though thiamin degradation was different. This likely demonstrates that the cook value is affected by the source of thiamin present in the food. Many researchers have identified different z-values for thiamin depending on the food undergoing thermal processing. For example, thiamin z-values were between 25.3°C and 26.3°C for periwinkle processed in various media [46], 31.4°C for tuna [47], and 26°C for rainbow trout [48]. As such, cook values calculated as done in the present experiment may not reflect the true nutritional quality of canned cat food when different thiamin sources are used. Other researchers have proposed 150–200 minutes as maximum cook values for thermally processed products [49]. However, there are no published standards for maximum cook values for pet foods. Instead, the common metric is whether or not the pet food meets the regulatory nutritional recommendations throughout its shelf-life. This was not addressed in the present experiment and highlights an area for further research.

Thiamin loss was not minimized by including the vitamin premix in the formula. This was not expected and suggested that supplemental thiamin mononitrate and thiamin intrinsic to standard canned cat food ingredients have similar retention due to thermal processing. However, the different yeasts tested did not exhibit similar thiamin degradation. The NY inclusion of the yeast fixed factor represented no yeast inclusion in the formula and quantified changes in thiamin present in standard canned cat food ingredients. The only yeast to exhibit similar changes in thiamin loss was BY. Even though LBV contained more thiamin than BY and EA, it exhibited the lowest retention. It is important to note that experimental treatments containing LBV met minimum recommended nutrient allowances [9] and were practical formulations

despite the higher thiamin degradation. Thiamin loss was not minimized by the combination of the vitamin premix and yeast. Instead, most formulations were similar and loss increased when vitamin premix was included with EA. All formulas were comparable to the formula containing the vitamin premix and no yeast, which was considered most similar to what is found in the marketplace, in terms of thiamin loss due to thermal processing.

It appeared thiamin intrinsic to BY had the most favorable thermal processing survival of the three yeasts tested. The ingredient contained more DMB thiamin than standard canned cat food ingredients (with the exception of vitamin premix), yet its thiamin loss was similar. Additional testing with this ingredient should determine the ideal amount of BY when batter viscosity, processed formula texture, and feline thiamin bioavailability are considered in addition to thiamin loss. The scheduled process employed in such an experiment should more closely resemble standard processing conditions instead of mimicking a worst-case scenario. Another area of further research could be to identify the mechanism behind the improved thiamin retention exhibited by BY. The lack of agreement between thiamin retention and cook values in the present experiment suggest that thiamin intrinsic to the different ingredients had varying levels of thermal resilience. However, investigation into potential mechanisms fell outside the scope of the present experiment. While the intention of this experiment was to evaluate yeast ingredients based on thiamin loss due to retort processing, it is clear that LBV had the highest thiamin content of any yeast examined. Therefore, it is reasonable to conduct further research with the ingredient even though it exhibited poor retention in the present experiment.

## Conclusion

The results from this experiment suggested that some yeast ingredients contain meaningful levels of thiamin to serve as sources in canned cat food. At the same time, thiamin degradation was different when three commercially available yeasts or no yeast were included in the diet formulation and the inclusion of a standard vitamin premix did not improve thiamin retention as expected. Formulations with LBV or BY alone and LBV, BY, or EA with the vitamin premix did result in products that met or exceeded nutrient allowances. They may be suitable sources of thiamin due to their high thiamin content relative to standard ingredients in commercial canned cat foods alone. The results discussed suggest that BY was the most favorable yeast in terms of thiamin loss due to thermal processing. Future research is necessary to fine-tune canned cat food formulas with this ingredient to meet all requirements for food safety, pet health, and pet owner acceptance. Such research could include storage of the ingredients alone and in processed diets to further describe the stability of their thiamin. Additionally, the lack of agreement between the calculated cook values and the observed thiamin degradation due to thermal processing indicate that thiamin supplied by different ingredients do not have the same z-value. Research identifying the z-value for these ingredients would result in more accurate cook values calculated with the equation used in the present experiment.

## Supporting information

**S1 Table. Nutritional composition of ingredients[1] used to produce canned cat food containing different sources of thiamin.**
(DOCX)

**S2 Table. Dry matter basis macronutrient and mineral contents (mean ± standard deviation) of pre-retort and post-retort canned cat foods containing different levels of a vitamin premix and/or a yeast ingredient[1].**
(DOCX)

**S1 File. Experimental data.**
(XLSX)

## Author Contributions

**Conceptualization:** Amanda N. Dainton.

**Data curation:** Amanda N. Dainton.

**Formal analysis:** Amanda N. Dainton.

**Investigation:** Amanda N. Dainton.

**Methodology:** Amanda N. Dainton, Markus F. Miller, Jr., Brittany White, Leah Lambrakis, Charles Gregory Aldrich.

**Project administration:** Charles Gregory Aldrich.

**Resources:** Amanda N. Dainton, Markus F. Miller, Jr., Brittany White, Leah Lambrakis, Charles Gregory Aldrich.

**Supervision:** Charles Gregory Aldrich.

**Visualization:** Amanda N. Dainton.

**Writing – original draft:** Amanda N. Dainton.

**Writing – review & editing:** Amanda N. Dainton, Markus F. Miller, Jr., Brittany White, Leah Lambrakis, Charles Gregory Aldrich.

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
