## [Decision Letter · Decision Letter 0]

26 Apr 2022

PONE-D-22-05056Composition and thermal processing evaluation of yeast ingredients as thiamin sources compared to a standard vitamin premix for canned cat foodPLOS ONE

Dear Dr. Aldrich,

Thank you for submitting your manuscript to PLOS ONE. After careful consideration, we feel that it has merit but does not fully meet PLOS ONE’s publication criteria as it currently stands. Therefore, we invite you to submit a revised version of the manuscript that addresses the points raised during the review process. I AGREE WITH BOTH REVIEWERS THAT THERE ARE USEFUL DATA IN THIS BODY OF WORK. HOWEVER, AUTHORS NEED TO ADDRESS THE VARIOUS COMMENTS CAREFULLY AND REVISE THE PAPER ACCORDINGLY. IF SOME OF THE DATA REPORTED DO NOT DIRECTLY ADDRESS THE STATED OBJECTIVES, THEN I SUGGEST TP MOVE IT TO A SUPPLEMENTAL FILE. THAT SHOULD HELP STREAMLINE THE PAPER.

We look forward to receiving your revised manuscript.

Kind regards,

Juan J Loor

Academic Editor

PLOS ONE

Journal Requirements:

(Ingredient and analytical costs were supported by Simmons Pet Food, Inc. They assisted with the study design, data collection, and preparation of the manuscript but imposed no restrictions.)

3. Thank you for stating the following in the Competing Interests:

We note that one or more of the authors have an affiliation to the commercial funders of this research study : (I have read the journal's policy and the authors of this manuscript have the following competing interests: A.N.D. was employed by Simmons Pet Food Inc. as a paid intern prior to conducting this research project; M.F.M. Jr., B.W., and L.L. were employed by Simmons Pet Food while this research project was conducted; C.G.A. has no competing interests to declare.)

Within your Competing Interests Statement, please confirm that this commercial affiliation does not alter your adherence to all PLOS ONE policies on sharing data and materials by including the following statement: ""This does not alter our adherence to  PLOS ONE policies on sharing data and materials.” (as detailed online in our guide for authors http://journals.plos.org/plosone/s/competing-interests). If this adherence statement is not accurate and  there are restrictions on sharing of data and/or materials, please state these. Please note that we cannot proceed with consideration of your article until this information has been declared.

Reviewers' comments:

Reviewer's Responses to Questions

**Comments to the Author**

1. Is the manuscript technically sound, and do the data support the conclusions?

Reviewer #1: Partly

Reviewer #2: Yes

2. Has the statistical analysis been performed appropriately and rigorously? 

Reviewer #1: Yes

Reviewer #2: Yes

3. Have the authors made all data underlying the findings in their manuscript fully available?

Reviewer #1: Yes

Reviewer #2: Yes

4. Is the manuscript presented in an intelligible fashion and written in standard English?

Reviewer #1: Yes

Reviewer #2: Yes

5. Review Comments to the Author

Reviewer #1: PONE-D-22-05056 Composition and thermal processing evaluation of yeast ingredients as thiamin sources compared to a standard vitamin premix for canned cat food

I found the manuscript to be thorough and the procedures well conducted. I think there is some useful information here but I cannot recommend acceptance of the manuscript in its present form. There are a number of issues relative to the data presented and the clarity of interpretation that should be clarified before resubmission.

Line

64 “the first definition is” I found this confusing in that there is really only one definition. Please reword for clarity.

68 Delete “the form”

69 Suggest thiamin in different forms.

85 You describe 6 yeasts, then in Table 2 show their composition. I have two isses; 1) if you want to include all of their compositions then it should be Table 1. 2) Somewhere you chose 3 of the 6 to include in the foods? I could never understand where or why you chose the yeasts you used?

Tables 3-7 There is a lot of info presented that I am not sure all has much value. I would suggest provide the pre-retort values (Table 5) and then in the text address any nutrients that were affected by retort. Because it is the premise of the paper, thiamin could be dealt with in a separate table where all values could be viewed simultaneously. Other data if desired could be added as an appendix where it could be qavailable to others but is really not a part of the results.

240 As presented, the outcomes are confounded by design. To report a change in thiamin concentration is inappropriate. The outcome is predetermined by the thiamin content which was not equal. I suggest you express these as the fraction surviving retort. I think this would be more meaningful.

277-342 I do not see the point of this discussion as it pertains to the objectives of the paper.

Reviewer #2: Yeast ingredients are examined in this manuscript as possible sources of thiamin in canned cat food. The purpose of this contribution is to clarify the potential use of an alternative source of thiamin for canned cat food in the pet food industry. There is good writing throughout and the experiment has some merit.

Even though it is an important topic, the study fails to present a scenario that is similar to an industrial production line. As, diets failed to achieve the minimum recommended by AAFCO; the humidity of the diets was not similar to the humidity found in commercial products, and the processing time was longer than what would be expected in an industrial setting.

Due to the effect that moisture content has on heat penetration, producing a diet with a higher moisture content and processing time compared to market-standard diets could lead to a greater degradation of thiamine than would be seen in daily practice.

Further, the authors mention that sulfites can lead to thiamine degradation but fail to mention whether this is an expected industry practice or if only a minority of products include sulfites. Accordingly, if sulfites are used as an industry practice, the results of this study do not correspond to a commercially expected outcome.

Furthermore, the study does not provide relevant information about the ingredients used. Although the study focused on the level of thiamine in the diet and its degradation, it is evident that other characteristics of the ingredient can negatively affect the quality of the diet as well. During the discussion, the authors discuss all the points that would need to be reviewed, considering that before being accepted for publication, it will be recommended to analyze the ingredients in terms of their amino acid characteristics, their minerals, and their fiber content (soluble and insoluble). Further, the use of crude fiber constrains the scientific discussion, because it has been noted that it is not a common method for measuring fiber in different industries and that many studies have shown that it is also not the ideal method for testing fiber in pet food and ingredients.

The discussion is also repetitive and with little explanation of the reason for the differences found between the ingredients, with many paragraphs listing the fact that the vitamin premix was not effective in preventing the loss of thiamine.

In spite of being a matter of interest, the authors fail to answer questions that are relevant to industrial production.

6. PLOS authors have the option to publish the peer review history of their article (what does this mean?). If published, this will include your full peer review and any attached files.

Reviewer #1: No

Reviewer #2: No

---

## [Author Response · Author response to Decision Letter 0]

9 Jun 2022

Dr. Juan J. Loor, Academic Editor, PLOS ONE:

Dear Dr. Loor,

Thank you for serving as the Academic Editor for our manuscript titled “Composition and thermal processing evaluation of yeast ingredients as thiamin sources compared to a standard vitamin premix for canned cat food” and for allowing us the opportunity to revise it. We acknowledge that you and the reviewers have given significant amounts of time and effort to share critical feedback to strengthen our submission. We have considered all comments made by the reviewers and incorporated changed into the manuscript. All changes are shown using track changes in the file titled “Revised Manuscript with Track Changes.” We have also provided a “clean” version titled “Revised manuscript” and updated the supplemental data file titled “S1_File” to correct a data entry error and the cover letter file titled “Cover letter.” We believe the changes we have incorporated into our manuscript further clarify the importance of the experiment and the rationale behind the experimental design and execution.

We have outlined a response to the Journal Requirements and to each of the reviewers’ comments below:

Journal Requirements

Comment 1. When submitting your revision, we need you to address these additional requirements.

Response: Thank you for reminding us of PLOS ONE’s style requirements. We have removed the bolding of the title and removed the short title completely.

Comment 2. Thank you for stating in your Funding Statement:

(Ingredient and analytical costs were supported by Simmons Pet Food, Inc. They assisted with the study design, data collection, and preparation of the manuscript but imposed no restrictions.)

Response: We received no additional external funding for this study and have amended our Funding Statement within our cover letter.

Comment 3. Thank you for stating the following in the Competing Interests:

We note that one or more of the authors have an affiliation to the commercial funders of this research study : (I have read the journal's policy and the authors of this manuscript have the following competing interests: A.N.D. was employed by Simmons Pet Food Inc. as a paid intern prior to conducting this research project; M.F.M. Jr., B.W., and L.L. were employed by Simmons Pet Food while this research project was conducted; C.G.A. has no competing interests to declare.)

Within your Competing Interests Statement, please confirm that this commercial affiliation does not alter your adherence to all PLOS ONE policies on sharing data and materials by including the following statement: ""This does not alter our adherence to PLOS ONE policies on sharing data and materials.” (as detailed online in our guide for authors http://journals.plos.org/plosone/s/competing- interests). If this adherence statement is not accurate and there are restrictions on sharing of data and/or materials, please state these. Please note that we cannot proceed with consideration of your article until this information has been declared.

Response: Thank you for clarifying the requirements for the Funding Statement and Competing Interests Statement. We have updated both statements and included them in our revised cover letter.

Comment 4. Please include captions for your Supporting Information files at the end of your manuscript, and update any in-text citations to match accordingly. Please see our Supporting Information guidelines for more information: http://journals.plos.org/plosone/s/supporting-information.

Response: Thank you for pointing out that we were missing a caption for our Supporting Information file. We have added this to the manuscript.

Comments from Reviewer 1

Comment 1. I found the manuscript to be thorough and the procedures well conducted. I think there is some useful information here but I cannot recommend acceptance of the manuscript in its present form. There are a number of issues relative to the data presented and the clarity of interpretation that should be clarified before resubmission.

Response: Thank you for providing us with valuable insights and suggestions. We feel your comments have further strengthened our manuscript.

Comment 2. Line 64: “the first definition is” I found this confusing in that there is really only one definition. Please reword for clarity.

Response: Thank you for bringing this to our attention. We have reworded this sentence to explicitly say that both categories of microorganisms and spores must be taken into account when processing canned pet foods.

Comment 3. Line 68: Delete “the form”

Response: Thank you for this suggestion. We have deleted “the form”.

Comment 4. Line 69: Suggest thiamin in different forms.

Response: We have taken this suggestion and replaced “thiamin in a different form” with “thiamin in different forms”.

Comment 5. Line 85: You describe 6 yeasts, then in Table 2 show their composition. I have two issues; 1) if you want to include all of their compositions then it should be in Table 1. 2) Somewhere you chose 3 of the 6 to include in the foods? I could never understand where or why you chose the yeasts you iused?

Response: Thank you for highlighting this area of confusion. We have switched the numbering of Table 1 and Table 2 and moved Table 1 (containing the nutritional composition of the yeast ingredients) into the Materials and Methods. The three yeasts included in the foods were chosen based on their thiamin content. Yeasts with higher thiamin contents were preferred as they were more likely to meet the minimum recommended thiamin content in the canned cat food. This is described in lines 105-109.

Comment 6. Tables 3-7: There is a lot of info presented that I am not sure all has much value. I would suggest provide the pre-retort values (Table 5) and then in the text address any nutrients that were affected by retort. Because it is the premise of the paper, thiamin could be dealt with in a separate table where all values could be viewed simultaneously. Other data if desired could be added as an appendix where it could be qavailable to others but is really not a part of the results.

Response: We appreciate your comments to improve the conciseness of our paper. We feel the information presented in these tables is valuable for readers, especially pet food formulators. However, we have combined the diet pre- and post-retort moisture and thiamin values into one table (Table 3) and moved all other information into Supplementary Tables. All in-text references to the tables have also been updated.

Comment 7. Line 240: As presented, the outcomes are confounded by design. To report a change in thiamin concentration is inappropriate. The outcome is predetermined by the thiamin content which was not equal. I suggest you express these as the fraction surviving retort. I think this would be more meaningful.

Response: We appreciate your concerns with how our thiamin degradation data is presented. Prior to submitting this manuscript, we consulted with a statistician who advised the data be analyzed as change in thiamin concentration with pre-retort thiamin as a covariate to account for the confounding effect instead of analyzing as the fraction surviving retort. However, we acknowledge that describing the data as you suggest may be easier for our readers to understand. We have added a sentence ranking the 8 diets from least degradation to greatest degradation and expressed the values as relative percentages (lines 319-324).

Comment 8. Lines 277-342: I do not see the point of this discussion as it pertains to the objectives of the paper.

Response: Thank you for highlighting this as an area in need of refinement. We felt this section of the discussion was important to the paper because the yeast ingredients supply more than thiamin to the nutritional content of the diet. However, we concede that this section is lengthy and have condensed it to better reflect the objectives of the paper.

Comments from Reviewer 2

Comment 1. Yeast ingredients are examined in this manuscript as possible sources of thiamin in canned cat food. The purpose of this contribution is to clarify the potential use of an alternative source of thiamin for canned cat food in the pet food industry. There is good writing throughout and the experiment has some merit.

Response: Thank you for providing these thorough and insightful comments about our manuscript.

Comment 2. Even though it is an important topic, the study fails to present a scenario that is similar to an industrial production line. As, diets failed to achieve the minimum recommended by AAFCO; the humidity of the diets was not similar to the humidity found in commercial products, and the processing time was longer than what would be expected in an industrial setting.

Due to the effect that moisture content has on heat penetration, producing a diet with a higher moisture content and processing time compared to market-standard diets could lead to a greater degradation of thiamine than would be seen in daily practice.

Response: Respectfully, we disagree with your comment that the study fails to present a scenario that is similar to an industrial production line. The diets were produced in one of the largest commercial canned pet food facilities in the world with a process that mimicked normal procedures as closely as possible.

Only two diets failed to meet an AAFCO minimum nutrient requirement measured in this experiment. The diets in question (no vitamin premix with either NY or EA) fell below the AAFCO minimum thiamin content for cats. This information is still highly relevant to the pet food industry as it helps research and development scientists decide which ingredients are most helpful in providing thiamin.

We conceded that moisture content of the diets produced in this experiment was higher than the typical pâté-style canned pet food in lines 517-519 and that moisture content is known to influence the rate of heat penetration during processing in lines 524-525. However, moisture contents were similar across the diets and we do not feel moisture was a confounding factor in the experiment. Additionally, commercial canned pet foods that consist of a meat chunk in a liquid (colloquially called “chunks in gravy”) may contain 82% moisture, which is very similar to the moisture content of diets in the present experiment. Language clarifying this point has been added in lines 519-523.

We conceded that the retort processing parameters resulted in greater processing than what is reported as typical for the pet food industry in lines 559-561. However, we also described that the retort processing parameters used were designed to mimic a worst-case scenario in production in lines 165-169. As such, we feel that our experiment is relevant because not every production is typical and situations do arise when a food is processed to a greater degree than intended. We did suggest that a follow-up experiment be conducted with processing parameters closer to the “typical” production in lines 562-564. However, this is outside the scope of the present experiment.

Comment 3. Further, the authors mention that sulfites can lead to thiamine degradation but fail to mention whether this is an expected industry practice or if only a minority of products include sulfites. Accordingly, if sulfites are used as an industry practice, the results of this study do not correspond to a commercially expected outcome.

Response: We appreciate that you’ve identified the presence of sulfites as a cause for degradation. However, in our laboratory’s experience, one would have to intentionally add sulfites in order to demonstrate an effect. While we are pointing to the range of potential factors, evaluating these factors was outside the scope of the experiment. For more details, one would encourage the reviewer to read the thesis of DeNoya, 2016.

Comment 4. Furthermore, the study does not provide relevant information about the ingredients used. Although the study focused on the level of thiamine in the diet and its degradation, it is evident that other characteristics of the ingredient can negatively affect the quality of the diet as well. During the discussion, the authors discuss all the points that would need to be reviewed, considering that before being accepted for publication, it will be recommended to analyze the ingredients in terms of their amino acid characteristics, their minerals, and their fiber content (soluble and insoluble). Further, the use of crude fiber constrains the scientific discussion, because it has been noted that it is not a common method for measuring fiber in different industries and that many studies have shown that it is also not the ideal method for testing fiber in pet food and ingredients.

Response: We appreciate the reviewer’s comments on the importance of additional nutritional information regarding the ingredients presented in the manuscript. Out of consideration for Reviewer 1, we have consolidated this section and moved the majority of this information to Supplementary Files. We concede that the additional nutritional information suggested would be valuable, however this fell outside the scope of the present experiment.

Comment 5. The discussion is also repetitive and with little explanation of the reason for the differences found between the ingredients, with many paragraphs listing the fact that the vitamin premix was not effective in preventing the loss of thiamine.

In spite of being a matter of interest, the authors fail to answer questions that are relevant to industrial production.

Response: Thank you for highlighting that our discussion of why we felt the inclusion of different ingredients resulted in differences could be clearer. We proposed that thiamin supplied by the different ingredients did not have similar z-values in lines 641-644 of the conclusion. We have added this to the discussion (lines 620-624) and concede that this could be confirmed in future experiments that fell outside the scope of the present manuscript.

---

## [Decision Letter · Decision Letter 1]

4 Jul 2022

Composition and thermal processing evaluation of yeast ingredients as thiamin sources compared to a standard vitamin premix for canned cat food

PONE-D-22-05056R1

Dear Dr. Aldrich,

We’re pleased to inform you that your manuscript has been judged scientifically suitable for publication and will be formally accepted for publication once it meets all outstanding technical requirements.

Kind regards,

Juan J Loor

Academic Editor

PLOS ONE

Additional Editor Comments (optional):

Reviewers' comments:

Reviewer's Responses to Questions

**Comments to the Author**

1. If the authors have adequately addressed your comments raised in a previous round of review and you feel that this manuscript is now acceptable for publication, you may indicate that here to bypass the “Comments to the Author” section, enter your conflict of interest statement in the “Confidential to Editor” section, and submit your "Accept" recommendation.

Reviewer #1: All comments have been addressed

2. Is the manuscript technically sound, and do the data support the conclusions?

Reviewer #1: Yes

3. Has the statistical analysis been performed appropriately and rigorously? 

Reviewer #1: Yes

4. Have the authors made all data underlying the findings in their manuscript fully available?

Reviewer #1: Yes

5. Is the manuscript presented in an intelligible fashion and written in standard English?

Reviewer #1: Yes

6. Review Comments to the Author

Reviewer #1: (No Response)

7. PLOS authors have the option to publish the peer review history of their article (what does this mean?). If published, this will include your full peer review and any attached files.

Reviewer #1: No

---

## [Editor Report · Acceptance letter]

22 Jul 2022

PONE-D-22-05056R1 

Composition and thermal processing evaluation of yeast ingredients as thiamin sources compared to a standard vitamin premix for canned cat food 

Dear Dr. Aldrich:

I'm pleased to inform you that your manuscript has been deemed suitable for publication in PLOS ONE. Congratulations! Your manuscript is now with our production department. 

Kind regards, 

on behalf of

Dr. Juan J Loor 

Academic Editor

PLOS ONE